# View-Invariant Policy Learning
# via Zero-Shot Novel View Synthesis

**Stephen Tian**[1], **Blake Wulfe**[2], **Kyle Sargent**[1],
**Katherine Liu**[2], **Sergey Zakharov**[2], **Vitor Guizilini**[2], **Jiajun Wu**[1]
[1]Stanford University  [2]Toyota Research Institute

**Abstract:** Large-scale visuomotor policy learning is a promising approach toward developing generalizable manipulation systems. Yet, policies that can be deployed on diverse embodiments, environments, and observational modalities remain elusive. In this work, we investigate how knowledge from large-scale visual data of the world may be used to address one axis of variation for generalizable manipulation: observational viewpoint. Specifically, we study single-image novel view synthesis models, which learn 3D-aware scene-level priors by rendering images of the same scene from alternate camera viewpoints given a single input image. For practical application to diverse robotic data, these models must operate *zero-shot*, performing view synthesis on unseen tasks and environments. We empirically analyze view synthesis models within a simple data-augmentation scheme that we call View Synthesis Augmentation (VISTA) to understand their capabilities for learning viewpoint-invariant policies from single-viewpoint demonstration data. Upon evaluating the robustness of policies trained with our method to out-of-distribution camera viewpoints, we find that they outperform baselines in both simulated and real-world manipulation tasks. Videos and additional visualizations are available at https://s-tian.github.io/projects/vista.

**Keywords:** generalization, visual imitation learning, view synthesis

## 1 Introduction

A foundation model for robotic manipulation must be able to perform a multitude of tasks, generalizing not only to different environments and goal specifications but also to varying robotic *embodiments*. A particular robotic embodiment often comes with its own sensor configuration and perception pipeline. This variety is a major challenge for current systems, which are often trained and deployed with carefully controlled or meticulously calibrated perception pipelines. One approach to training models that can scale to diverse tasks as well as perceptual inputs is to train on a common modality, such as third-person RGB images, for which diverse data are relatively plentiful [1].

A challenge in using these data is that policies learned by current methods struggle to generalize across perceptual shifts for single RGB images. In

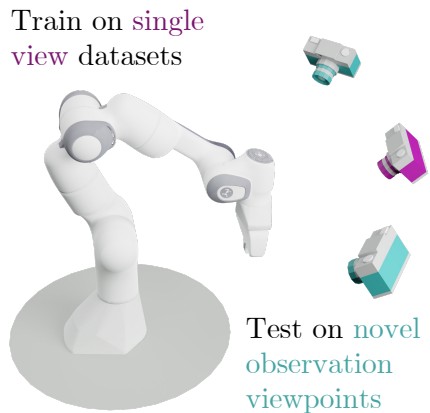

Figure 1: We aim to learn policies that generalize to novel viewpoints from widely available, offline single-view RGB robotic trajectory data.

this paper, we study one ubiquitous and practically challenging shift: when the camera viewpoint is altered. Prior studies have found that policies trained on RGB images collected from fixed viewpoints are consistently unable to generalize to visual inputs from other camera poses [2, 3, 4].

8th Conference on Robot Learning (CoRL 2024), Munich, Germany.

Existing approaches to learning viewpoint invariance include training using augmented data collected at scale in simulation [5, 6] or physically varying camera poses when collecting large-scale real robot datasets [7]. However, these strategies require resolving the additional challenges of sim-to-real transfer and significant manual human effort, respectively.

In this work, we leverage the insight that 3D priors can be obtained by generative models from large-scale (potentially robot-free) data and used to make robot policies more robust to changes in camera pose. We take a simple data augmentation approach to this problem by sampling views from a 3D-aware image diffusion novel view synthesis (NVS) model during policy training time. In training on these augmented views, the policy becomes robust to images from out-of-distribution camera viewpoints. We refer to this approach as **View Synthesis Augmentation (VISTA)**.

VISTA has several advantages. First, it can leverage large-scale 2D image datasets, which are more diverse than existing robotic interaction datasets with explicit 3D observations. Second, if in-domain robotic data *is* available, performance may be further improved via finetuning. Third, neither depth information nor camera calibration is required. Fourth, no limitations are placed on the form of the policy. While we focus on imitation learning, VISTA can also be applied to other robotic learning paradigms. Lastly, policy inference time is not impacted, as we do not modify inference behavior.

We first investigate the performance of a diffusion-based novel view synthesis model, ZeroNVS [8], when applied using our VISTA data augmentation scheme, and perform an empirical analysis of its performance with respect to various viewpoint distributions. Then, we investigate how finetuning an NVS model with in-domain data of robotic tasks can improve downstream policy robustness for held-out tasks. Finally, we show that these models can be used to learn viewpoint-robust policies from real robotic datasets. We demonstrate the potential for NVS models trained on large diverse robotic data to provide these priors across robot tasks and environments, finding that finetuning ZeroNVS models on the DROID dataset [7] can improve downstream real-world policy performance.

## 2 Related Work

**Learning viewpoint-robust robotic policies.** Learning deep neural network policies that can generalize to different observational viewpoints has been discussed at length in the literature. One set of approaches effectively augment the input data to a learned policy or dynamics model with additional 2D images rendered from differing camera viewpoints. These renderings are often obtained from simulators [5, 6] or by reprojecting 2D images [9]. Augmenting training with simulator data can improve robustness on simulation environments, but these methods must then address the challenge of sim-to-real transfer for deployment on real systems. In this work, we study methods for learning invariant policies directly using robot data from the deployment setting, including real robot trajectories. Existing work [10] performs view augmentation of real-world wrist camera images; however, this is performed with the goal of mitigating covariate shift as opposed to improving camera pose robustness, and requires many views of a static scene to generate realistic novel views.

Another line of work forms explicit 3D representations of the scene such as point clouds or voxels to leverage equivariance properties [11, 12, 13, 14, 15], or projects from these representations to 2D images originating from canonical camera poses [16]. While these approaches have been shown to be robust to novel camera views [3], they require well-calibrated camera extrinsics, which can be practically challenging and time-consuming to collect, and are not present in all existing datasets (for example, the majority of datasets in Open X-Embodiment [1] do not provide camera extrinsics).

Rather than rely on explicit 3D representations, a related body of work learns latent representations that are robust to variations in camera pose. These methods often use view synthesis or contrastive learning as a pretraining or auxiliary objective [17, 18, 19, 6], and also often require accurate extrinsics, can be computationally expensive to run at inference time, or impose restrictive requirements on the latent space that can make multi-task learning challenging.

A technique that has shown promise in reducing the observational domain gap in robotic learning settings is the use of wrist-mounted or eye-in-hand cameras as part of the observation space [20, 21].

However, this does not obviate the need for third-person observations as it only provides information local to the gripper. We corroborate in our experiments that wrist-mounted camera observations are helpful but not solely sufficient for learning viewpoint-robust policies, and further that the use of wrist cameras can yield improvements orthogonal to the use of augmentation for third-person views.

**Single-image novel view synthesis.** Single-image novel view synthesis methods aim to reconstruct an image from a target viewpoint given a single image from a source viewpoint. One set of methods for novel view synthesis infers neural radiance fields from one or a few images [22, 23]. Another recent line of work trains diffusion models on images to perform novel view synthesis, and then distills 3D representations from these models [24]. These approaches have been extended to scene-level view synthesis [8, 25, 26], making them amenable to robotic manipulation settings. They have been largely developed, trained, and evaluated on large video datasets; however, to our knowledge, their application in robotic policy learning remains relatively unexplored.

**Generative image models in robotics.** Pretrained image generation models have been applied in the context of robotic manipulation via *semantic* data augmentation [27, 28], where the goal is for the policy to better generalize to unseen backgrounds or objects as opposed to camera viewpoints. Similar generative models have also been applied to improve cross-embodiment transfer of policies [29] and as high-level planners using an image subgoal interface [30, 31, 32, 33, 34]. Overall, these methods address different challenges and are largely complementary to our method.

# 3 Preliminaries

## 3.1 Problem Statement

The techniques we discuss can be flexibly applied to many visuomotor policy learning settings; however, for a systematic and computationally constrained evaluation, we choose to study them in the context of visual imitation learning.

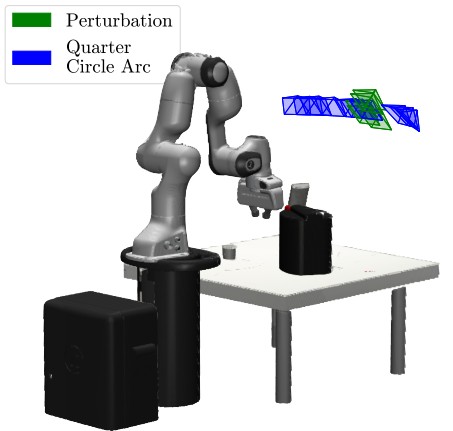

We frame each robotic manipulation problem as a discrete-time partially observed Markov decision process (POMDP), with state space $\mathcal{S}$, action space $\mathcal{A}$, transition function $P$, reward function $R$, and observation function $\mathcal{O}$. This observation function maps states and actions into the observation space conditioned on extrinsic parameters

Figure 2: Random samples from the two considered evaluation viewpoint ranges.

$E$. We assume access to a dataset $\mathcal{D}$ consisting of $M$ expert demonstrations $\tau_{0:M}$: $\tau_i = \left\{(o_0, a_0, \ldots, o_t, a_t, \ldots, o_T, a_T)\right\}$ where $T$ is the total number of timesteps in a particular demonstration. Concretely, the observation $o$ consists of both low-dimensional observations in the form of robot proprioceptive information, as well as RGB image observations $o_I \in \mathbb{R}^{H \times W \times 3}$ captured by a *fixed* third-person camera with extrinsics $E_{\text{orig}}$.

The objective is to learn a policy $\pi(a|o)$ that solves the task, where observed images $o_I$ are captured by a camera with extrinsics $E_{\text{test}}$ sampled from a distribution $\mathcal{E}_{\text{test}}$. Critically, we do not assume access to the environment or the ability to place additional sensors at training time.

## 3.2 Zero-Shot Novel View Synthesis from a Single Image

We define the single-image novel view synthesis (NVS) problem as finding a function $\mathcal{M}(I_{\text{context}}, f, E_{\text{context}}, E_{\text{target}})$ that, given an input image $I_{\text{context}} \in \mathbb{R}^{H \times W \times 3}$ of a scene captured with camera extrinsics (e.g., camera pose) $E_{\text{context}}$ and simplified intrinsics represented by a field of view $f$, renders an image of the same scene $I_{\text{context}}$ captured with camera extrinsics $E_{\text{target}}$.

To extend this setting to *zero-shot* novel view synthesis, we further assume that the image $I_{\text{orig}}$ depicts a robotic task that was not seen when training $\mathcal{M}$. As we will describe in Section 5, we

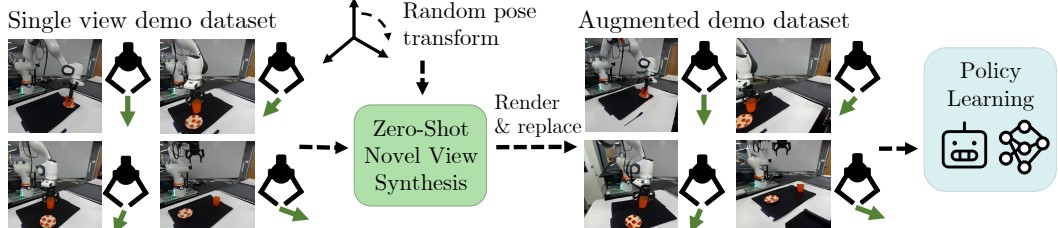

Figure 3: Depiction of the data augmentation scheme that we study. Observations are replaced with viewpoint-augmented versions of the same scene with action labels held constant.

conduct experiments on both models that have never seen robotic data, as well as models finetuned on robotic data from simulated pre-training tasks and large-scale, real-world data.

## 4  Learning View Invariance with Zero-Shot Novel View Synthesis Models

In this section, we describe VISTA, the data augmentation scheme for view-invariant policy learning that we study in the remainder of the paper. The method is summarized in Figure 3.

To learn viewpoint-invariance, some prior works augment experience with images rendered from virtual cameras in simulation [5, 6]. However, we wish to learn viewpoint-invariant policies directly from existing offline datasets, which could be from inaccessible simulated environments or data collected in the real world. Furthermore, many robotic datasets do not contain the multiview observations or depth images needed for 3D reconstruction. Thus, we explore using *single image* novel view synthesis methods to perform augmentation.

Concretely, given a single-image novel view synthesis model $\mathcal{M}(I_{\text{context}}, f, E_{\text{context}}, E_{\text{target}})$, VISTA uses $\mathcal{M}$ to replace *each frame* of a demonstration trajectory with a synthesized frame with independently randomly sampled target extrinsics $E_{\text{target}} \sim \mathcal{E}_{\text{train}}$. That is, we independently replace each observation-action pair $(o, a)$ with $(\mathcal{M}(o_I, f, E_{\text{context}}, E_{\text{target}}), a)$. For the sake of systematic evaluation, in our simulated experiments, we assume knowledge of both the initial camera pose $E_{\text{context}}$ and the target distribution $\mathcal{E}_{\text{target}}$. However, the novel view synthesis models we study use only the *relative poses* between $E_{\text{context}}$ and $E_{\text{target}}$; absolute poses are not required and are not used in real-world experiments. We assume that the field of view is known.

VISTA has several appealing properties. First, while methods that form explicit 3D representations must either use multi-view images or assume static scenes when performing structure-from-motion, it avoids the computational expense of 3D reconstruction and takes advantage of the fact that a scene is static at any slice in time. Second, VISTA does not add additional computational complexity at inference time, as the trained policy's forward pass remains the same. Lastly, VISTA inherits improvements in the modeling and generalization capability of novel view synthesis models.

We center our analysis around a particular novel-view synthesis model, ZeroNVS [8]. ZeroNVS is a latent diffusion model that generates novel views of an input image given a specified camera transformation. It is initialized from Stable Diffusion [35] and fine-tuned on a diverse collection of 3D scenes, therefore achieving strong zero-shot performance on a wide variety of scene types. Moreover, as a generative model, it tends to generate novel views which are crisp and realistic, mitigating the domain gap between generated and real images.

Although ZeroNVS provides reasonable predictions even in zero-shot settings, we found that it also has failure modes that generate images that appear to contain extreme close-ups of objects in the scene, potentially due to poor extrinsics in the training dataset. To partially address these scenarios, we simply reject and resample images that have a perceptual similarity (LPIPS) [36] distance larger than a value $\eta$ from the input image, which we found to slightly improve performance.

While many techniques for imitation learning have been proposed, as a strong baseline that fits our computational budget, we use the implementation of behavior cloning with a Gaussian mixture

| Task | Original | Reproj. | PixelNeRF | ZeroNVS | ZeroNVS (FT) | Sim (oracle) |
|------|----------|---------|-----------|---------|--------------|--------------|

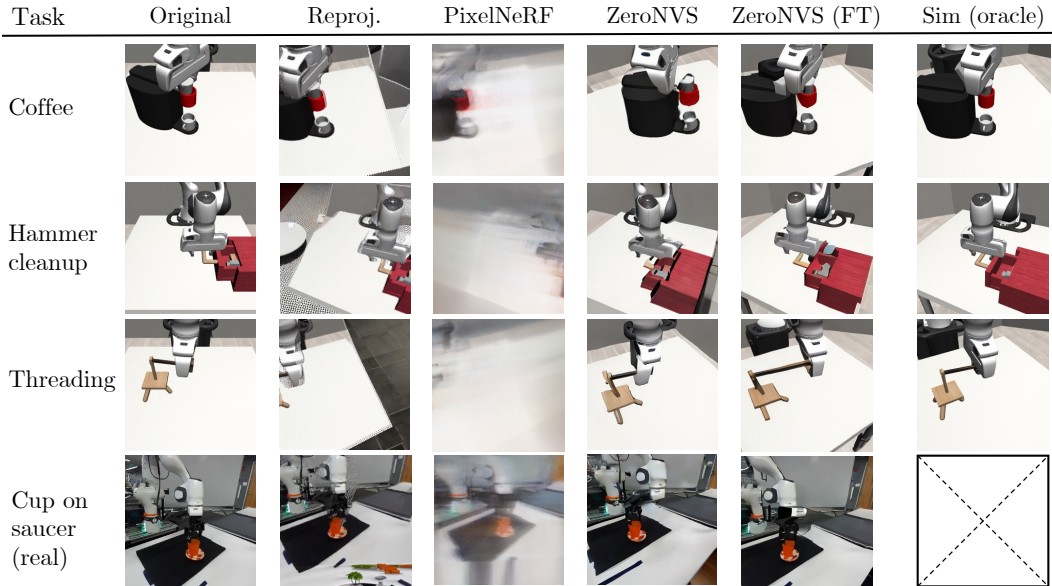

Figure 4: Qualitative examples of novel views rendered on robotic tasks. All images are synthesized *zero-shot*; that is, models have **not** been previously trained on data from that task. We observe that finetuning on robotic datasets improves image fidelity, particularly for robot appearances.

model output from robomimic [37] in our simulated experiments. In real-world experiments, we instead train diffusion policies [38] due to their success in learning policies for real robots.

For additional implementation details and pseudocode of VISTA, please see Appendix A.

## 5 Experimental Analysis

In this section, we perform empirical analyses to answer the following questions:

**Q1:** Can policies trained with data augmented by novel view synthesis models trained on large-scale out-of-domain datasets improve robustness to novel viewpoints? How do these models compare to existing alternatives?

**Q2:** Can finetuning novel view synthesis models on robotic data improve the performance of VISTA when applied to unseen tasks with larger viewpoint changes?

**Q3:** How do methods providing augmented third-person viewpoints interact with strategies for reducing the observational domain gap, such as adding wrist-mounted cameras?

**Q4:** Can VISTA be applied to learn policies on real robots directly using real-world data? How does finetuning view synthesis models on diverse real robot data affect downstream performance?

**Simulated experimental setup**. We perform simulated experiments using the robomimic framework built on the MuJoCo simulator [39], with additional tasks introduced in MimicGen [40]. For the Lift and Nut Assembly tasks, we use the proficient-human expert demonstration datasets from robomimic for training. For the remainder of the tasks, we train using the first 200 demonstrations of D0 datasets and evaluate using the D0 environment variants as defined by MimicGen.

To contextualize the results, we introduce the following baseline methods:

- **Single view.** To represent the performance of a model trained without view-invariance, this performs behavioral cloning using only the source demonstration dataset.
- **Simulator (oracle).** As an upper bound of the performance of per-frame random augmentation for learning viewpoint invariance, this baseline directly uses the simulator to render novel viewpoints.
- **Depth estimation + reprojection.** This baseline follows the augmentation scheme described in Section 4. It synthesizes novel views from RGB images using a three-stage pipeline. Because we do not assume access to depth, it first performs metric depth estimation using an off-the-shelf

| Aug. model | Lift | Threading | Nut Asm. |
|---|---|---|---|
| Single view | $72.7 \pm 3.3$ | $9.3 \pm 0.7$ | $16.7 \pm 1.8$ |
| Depth est. + Reproj. | $93.3 \pm 1.8$ | $12.0 \pm 1.2$ | $29.3 \pm 0.7$ |
| PixelNeRF [22] | $44.7 \pm 10.9$ | $4.7 \pm 0.7$ | $10.7 \pm 1.8$ |
| ZeroNVS [8] | $\mathbf{95.3 \pm 2.4}$ | $\mathbf{23.3 \pm 2.4}$ | $\mathbf{36.0 \pm 0.0}$ |
| Simulator (oracle) | $100.0 \pm 0.0$ | $53.3 \pm 1.8$ | $51.3 \pm 1.8$ |

Table 1: **Policy performance on perturbed viewpoints.** Policy success rates on randomized test viewpoints as percentages and standard error of the mean (SEM) over 3 random seeds when performing per-frame data augmentation with view synthesis methods. We report the maximum performance across training checkpoints, evaluating for 50 trials following Mandlekar et al. [37].

| | Unseen Object | | Shared Object | | X-Embodiment | |
|---|---|---|---|---|---|---|
| Aug. model | Threading | Hammer | Coffee | Stack | PickPlace | Nut Asm. |
| Single view | $10.0 \pm 1.2$ | $18.0 \pm 2.3$ | $10.0 \pm 1.2$ | $49.3 \pm 3.7$ | $31.3 \pm 0.7$ | $10.7 \pm 0.7$ |
| Depth est.+Reproj. | $7.3 \pm 1.3$ | $20.0 \pm 1.2$ | $9.3 \pm 2.4$ | $36.0 \pm 2.3$ | $28.7 \pm 0.7$ | $10.7 \pm 1.3$ |
| ZeroNVS [8] | $17.3 \pm 1.8$ | $27.3 \pm 3.7$ | $15.3 \pm 1.3$ | $52.7 \pm 2.4$ | $32.0 \pm 2.3$ | $18.7 \pm 0.7$ |
| ZeroNVS (MimicGen) | $\mathbf{32.0 \pm 0.0}$ | $\mathbf{52.0 \pm 3.5}$ | $\mathbf{32.7 \pm 2.4}$ | $\mathbf{61.3 \pm 2.4}$ | $\mathbf{40.7 \pm 3.5}$ | $\mathbf{26.0 \pm 2.0}$ |
| Simulator (oracle) | $60.7 \pm 0.7$ | $100.0 \pm 0.0$ | $84.0 \pm 2.0$ | $86.0 \pm 2.3$ | $90.0 \pm 2.3$ | $56.0 \pm 3.1$ |

Table 2: **Policy performance on quarter circle arc viewpoints.** We report success rates and standard error of the mean over 3 random seeds. Finetuning ZeroNVS on simulated robotic data significantly improves performance across all tasks in this setting.

model [41]. Next, it lifts the RGBD information into a 3D point cloud with a pinhole camera model and then renders the point cloud into an RGB image at the target camera extrinsics. Finally, because this reprojection often produces partial images, we perform inpainting of "holes" and outpainting of image boundaries using a pretrained diffusion model [35].

- **PixelNeRF.** To evaluate differences between novel view synthesis models, we evaluate a method that performs per-frame viewpoint augmentation using a PixelNeRF [22] model trained on the same mixture of datasets as ZeroNVS. PixelNeRF uses a convolutional encoder to condition a neural radiance field [42], which is then rendered from the novel viewpoint.

Further details regarding baseline implementations and hyperparameters are in Appendix A.

**Q1. Performance of pre-trained novel view synthesis models.** First, we seek to evaluate the performance of view synthesis models that rely on large-scale, diverse pretraining. We test a distribution of test camera poses denoted *perturbations*, that are representative of incremental changes, for instance, that of a physical camera drifting over time or subject to unintentional physical disturbance. Specifically, this distribution is parameterized by a random translation $\Delta t \sim \mathcal{N}(0, \text{diag}(\sigma_t^2))$ and rotation around a uniformly randomly sampled axis, where the magnitude is sampled from $\mathcal{N}(0, \sigma_r{}^2)$. Samples from this range are visualized in Figure 2, and example observations are in Appendix B.

The results are presented in Table 1. First, we note that the oracle simulator augmentation scheme is able to reclaim a significant portion of policy performance that is lost by only training on the original data (single view). We find that the depth estimation + reprojection method is able to consistently provide modest improvements to the performance on test viewpoints. Among the fully neural methods, PixelNeRF does not synthesize views with sufficient fidelity, and causes even a drop in performance compared to not doing augmentation. We thus omit this baseline in further evaluations. However, we find that a pretrained ZeroNVS model, *despite (likely) having never seen an image of a robotic arm during training*, is able to improve novel view performance even further.

**Q2. Effect of finetuning view synthesis models on in-domain data.** Next, we investigate whether finetuning these novel view synthesis models on in-domain data can yield improved performance when applied to unseen tasks. To test this, we study a more challenging distribution of camera poses with a real-world analogue to constructing another view of a given scene. We first compute a sphere centered at the robot base and containing the initial camera pose. We then sample camera poses on the sphere at the same $z$ height and within a $90°$ azimuthal angle of the starting viewpoint. The

radius of the sphere is further randomly perturbed with Gaussian noise with variance $\sigma_r^2$. We call this distribution *quarter circle arc*, with samples shown in Figure 2 and more details in Appendix B.

We finetune the ZeroNVS model on a multi-view dataset generated using eight MimicGen tasks: `stack three`, `square`, `three piece assembly`, `mug cleanup`, `pick place can`, `nut assembly`, `kitchen`, and `coffee prep`. Additional finetuning details can be found in Appendix B. We then test the model, denoted **ZeroNVS (MimicGen)**, when used for augmentation on datasets of held-out tasks, which we categorize below by their relationship with the finetuning tasks.

- **Unseen Object:** Tasks contain objects that are not present in any finetuning tasks.
- **Shared Object:** Tasks contain objects that are present in one or more finetuning tasks, but in the context of a set of different objects or scenes.
- **X-Embodiment:** The same task is present in the finetuning data, but is performed by a different robot (Rethink Sawyer instead of Franka Panda).

Quantitative results are presented in Table 2. In this more challenging setting, improvements from the depth estimation + reprojection baseline are much more limited, likely because many requested novel viewpoints are outside the original camera's viewing frustum. The pretrained ZeroNVS model yields modest improvements on all tasks. We see the best performance when using the model finetuned on the MimicGen data, often doubling the success rate of the next best method.

Qualitatively, as seen in Figure 4, we find that the ZeroNVS model finetuned on MimicGen data produces higher fidelity images, particularly with respect to the robotic arm's appearance.

**Q3. Use of wrist-mounted cameras to reduce domain gap.** Wrist-mounted cameras are a popular and effective approach to improving visuomotor policy performance and reducing domain shift due to changes in visual observations [20]. In this experiment, we examine the effect of using wrist-camera observations in conjunction with augmented third-person views. The results are shown in Figure 5. We see that adding wrist camera observations slightly improves performance on the threading task for all augmentation techniques, suggesting that methods for view-invariance for third-person views can be complementary to the use of wrist cameras. For the `threading` task, the performance of a policy using solely wrist observations, which are unperturbed at test time, is $58\%$. This is better than even our strongest policy using a third-person view augmentation model. However, the performance of wrist-camera-only policies may be limited for many tasks [20]. For instance, in `threading`, the oracle augmentation + wrist camera policy achieves a $73\%$ success rate using the original third-person viewpoint.

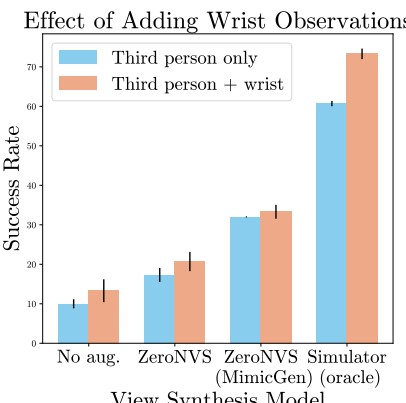

Figure 5: Performance of novel view–augmented policies when provided with additional wrist camera observations, which are consistent between train and test settings. We find as per expectation that wrist observations improve performance across the board, as they are agnostic to third-person viewpoint. These improvements complement those of view augmentation methods.

**Q4. Performance on real robots.** We further investigate the performance of VISTA when training policies on real-world data. Critically, we also seek to validate whether finetuning NVS models on large-scale real multi-view robotic data can improve performance for real-world policies.

To test this, we first finetune a ZeroNVS model on a subset of the DROID [7] dataset, which contains over 75k trajectories of a variety of tasks in diverse environments. We randomly sample a subset of 3000 trajectories in DROID and sample 10 random timesteps within each trajectory as "scenes" for finetuning, using the four external views from two stereo cameras as paired data. We then collect a dataset consisting of 150 expert demonstrations on a Franka Emika Panda robot for the task `place cup on saucer` from a single camera viewpoint. We train diffusion policies [38] on this data following the configuration in DROID [7] with four policy variants as follows: **Original**

| Original | Camera 2 | Camera 3 | Camera 4 | Camera 5 |
|---|---|---|---|---|
| 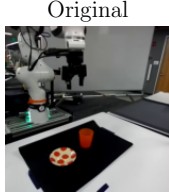 | 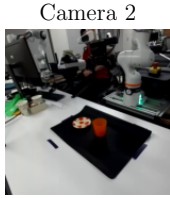 | 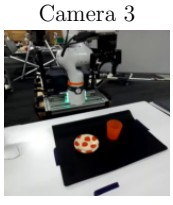 | 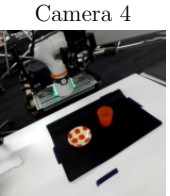 | 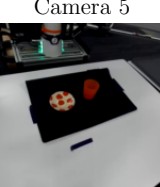 |

Figure 6: Tested camera viewpoints for real world experiments. We vary both the position and orientation of the cameras at a range of distances. Note that camera calibration information is not used for new viewpoints.

| | Orig. Cam | | Cam 2 | | Cam 3 | | Cam 4 | | Cam 5 | | Agg. | |
|---|---|---|---|---|---|---|---|---|---|---|---|---|
| Aug. model | R | C | R | C | R | C | R | C | R | C | R | C |
| Original data+wrist | 9/10 | 4/10 | 2/10 | 0/10 | 1/10 | 0/10 | 1/10 | 0/10 | – | – | 14/40 | 4/40 |
| ZeroNVS aug+wrist | 8/10 | 3/10 | 7/10 | 2/10 | 8/10 | 4/10 | 9/10 | 3/10 | 7/10 | 2/10 | 39/50 | 14/50 |
| ZeroNVS (DROID)+wrist | 9/10 | 6/10 | 8/10 | 2/10 | 9/10 | 5/10 | 9/10 | 3/10 | 9/10 | 4/10 | **44/50** | **20/50** |
| Wrist only | – | – | – | – | – | – | – | – | – | – | 16/20 | 5/20 |

Table 3: **Real robot policy success rates.** We evaluate each rollout's success on two stages of the task: **R**eaching the cup and positioning the gripper for a grasp as determined by a human rater, and **C**ompleting the full `place cup on saucer` task. Camera 5 results for "original data + wrist" are omitted as the policy exhibited qualitatively essentially random behaviors for novel views. We find that the ZeroNVS augmentation improves viewpoint robustness, and using the DROID-finetuned NVS model yields additional gains.

**data + wrist** uses the third-person camera view and wrist view as policy inputs. **ZeroNVS aug + wrist** additionally performs augmentation on the third-person view using the ZeroNVS model from the original paper. **ZeroNVS (DROID) + wrist** uses the NVS model finetuned on DROID data instead. Finally, representing an alternative approach that sidesteps the viewpoint shift problem entirely by only using the wrist camera, which is always fixed related to the end effector, **wrist only** is a baseline that does not use third-person camera inputs. We evaluate these policies when the external observations are captured from the original viewpoint and four novel views (see Figure 6). Full finetuning and real world experimental setup details are in Appendix C.

The results, presented in Table 3, indicate that VISTA is also effective in improving policy viewpoint robustness in the real world settings. Further, we see a performance gain from finetuning the NVS model on a large, diverse robotic dataset. In contrast, the policy trained on the single viewpoint data struggles to reliably reach toward the cup under viewpoint changes.

## 6 Conclusion and Limitations

**Limitations**. While VISTA is effective at improving the viewpoint robustness of policies, it does have certain limitations. First, the computational expense of generating novel views can be significant. Second, augmenting views during policy training can increase training time and therefore computational expense. Third, sampling views during data augmentation requires some distribution of poses from which to sample. This distribution must cover the reasonable space of views expected at deployment time. Fourth, single-image novel view synthesis models often perform poorly when the novel view is at a camera pose that differs dramatically from the original camera pose, and this limits the distribution from which views may be sampled during data augmentation.

**Conclusion.** In this paper, we presented VISTA, a simple yet effective method for making policies robust to changes in camera pose between training and deployment time. Using 3D priors from single image view synthesis methods trained on large-scale data, VISTA performs data augmentation to learn policies invariant to camera pose in an imitation learning context. Experiments in both simulated and real world environments demonstrated improved robustness to novel viewpoints of our approach over baselines, particularly when using view synthesis models finetuned on robotic data (though applied zero-shot with respect to tasks). There are a number of promising directions for future work, but of particular interest is studying the performance of this data augmentation scheme at scale across a large dataset of robotic demonstrations.

## Acknowledgments

We thank Hanh Nguyen and Patrick "Tree" Miller for their help with the real robot experiments, and Weiyu Liu, Kyle Hsu, Hong-Xing "Koven" Yu, Chen Geng, Ruohan Zhang, Josiah Wong, Chen Wang, Wenlong Huang, and the SVL PAIR group for helpful discussions. This work is in part supported by the Toyota Research Institute (TRI), NSF RI #2211258, and ONR MURI N00014-22-1-2740. ST is supported by NSF GRFP Grant No. DGE-1656518.

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

# A   Details of Model Implementations

All novel view synthesis methods that we consider generate novel views at a resolution of $256 \times 256$ given RGB images at a resolution of $256 \times 256$. The synthesized images are later downsampled for policy training. To clarify how these models are used in a policy learning pipeline, we provide pseudocode of the VISTA algorithm in Algorithm 1.

---

**Algorithm 1** VISTA: Learning View Invariant Policies via Novel View Synthesis.

---

**Require:** Dataset $\mathcal{D}$ consisting of trajectories $\tau_{0:M} : \tau_i = \{(o_0, a_0, \cdots, o_t, a_t, \cdots o_T, a_T)\}$, novel view synthesis model $\mathcal{M}(I_{context}, f, E_{context}, E_{target})$, training camera extrinsics distribution $\mathcal{E}_{train}$, LPIPS threshold $\eta$, number of generation attempts $M$, known camera field-of-view $f$, number of augmented frames per reference frame $C$, policy learning procedure LEARNPOLICY$(\mathcal{D})$.

1: $\mathcal{D}' \leftarrow \{\}$                                          ▷ *Initialize augmented dataset $\mathcal{D}'$.*
2: **for** trajectory $\tau_i$ in $\mathcal{D}$ **do**                       ▷ *Iterate over all transitions in source dataset.*
3:     **for** transition $(o_t, a_t)$ in $\tau_i$ **do**
4:         **for** each of $C$ augmented copies **do**
5:             $E_{target} \sim \mathcal{E}_{target}$            ▷ *Randomly sample extrinsics from training distribution.*
6:             $num\_tries \leftarrow 0$
7:             **repeat**
8:                 $o'_t \leftarrow \mathcal{M}(o_t, f, I, E_{target})$                      ▷ *Synthesize novel view image.*
9:                 $num\_tries \leftarrow num\_tries + 1$
10:            **until** LPIPS$(o'_t, o_t) < \eta$ or $num\_tries >= M$
11:            ▷ *Reject images that are too far away in LPIPS, or give up after too many failures.*
                  *Note that we only do this with the ZeroNVS model, to filter very poor generations.*
                  *With other models, all generated images are accepted, i.e, $\eta = \infty$.*                ◁
12:            **if** $num\_tries < M$ **then**                       ▷ *If novel novel synthesis was successful*
13:                $\mathcal{D}' \leftarrow \mathcal{D}' \cup (o'_t, a_t)$                       ▷ *Add augmented transition to buffer.*
14:            **else**
15:                $\mathcal{D}' \leftarrow \mathcal{D}' \cup (o_t, a_t)$                        ▷ *Add original transition to buffer.*
16: ▷ *Learn policy on augmented dataset, in our case, imitation learning via behavior cloning.*         ◁
17: $\pi \leftarrow$ LEARNPOLICY$(\mathcal{D}')$
18: **return** $\pi$

---

## A.1   ZeroNVS

ZeroNVS is a latent diffusion model that generates novel views of an input image given a specified camera transformation. It is initialized from Stable Diffusion and then fine-tuned on a diverse collection of 3D scenes, and therefore achieves strong zero-shot performance on a wide variety of scene types. Moreover, as a generative model, it tends to generate novel views which are crisp and realistic, mitigating the domain gap between generated and real images. This distinguishes ZeroNVS from methods such as PixelNeRF [22], which are trained with regression-based losses and tend to produce blurry novel views even for small camera motion.

We use the implementation and pretrained checkpoint provided by Sargent et al. [8]. As mentioned in Section A, although ZeroNVS largely produces reasonable views even zero-shot, it can sometimes produce images with significant visual artifacts. To filter these out, we reject and resample images that have a LPIPS [36] distance larger than a hyperparameter $\eta$ from the input image. We set $\eta = 0.5$ for all simulated experiments and $\eta = 0.7$ for real experiments. We do not extensively tune this hyperparameter. If the model fails to produce an image with distance $< \eta$ after 5 tries, the original image is returned.

ZeroNVS also requires as input a scene scale parameter. To determine the value of the scene scale for simulated experiments, we perform view synthesis using the pretrained ZeroNVS checkpoint on a set of 100 test trajectories for the `lift` and `threading` environments, and compute the LPIPS score between the ZeroNVS rendered images and ground truth simulator renders for values $\{0.4, 0.45, 0.5, 0.55, 0.6, 0.65, 0.7, 0.75, 0.8, 0.85, 0.9, 0.95\}$. We find that the lowest error across the tasks is achieved at $0.6$ and thus use $0.6$ for all environments, including the real robot experiments.

While the behavior of the ZeroNVS model is somewhat sensitive to scene scale, we believe this may be alleviated by selecting a wider viewpoint randomization radius at training time, which is corroborated by our real robot experiments.

When sampling, we perform $250$ DDIM steps and use a DDIM $\eta$ of $1.0$. We use a field of view (FOV) of $45$ degrees for simulated experiments (obtained from the simulator camera parameters) and FOV of $70$ degrees for the real world experiments (obtained from the Zed 2 camera datasheet).

Sampling the diffusion model for NVS is roughly similar to sampling from the vanilla Stable Diffusion model; it takes on average 8.7 seconds to generate a single $256 \times 256$ image with ZeroNVS using these settings on a single NVIDIA RTX 3090 GPU.

### A.2 Depth estimation + Reprojection baseline

This baseline represents a geometry-based approach that leverages depth estimation models trained on large-scale, diverse data.

First, we use ZoeDepth (ZoeD_NK) [41], an off-the-shelf model, to perform metric depth estimation on the input RGB image. Next, we deproject the images into pointclouds using a pinhole camera model. We rasterize an image from the points using the Pytorch3D point rasterizer [43], setting each point to have a radius of $0.007$ and $8$ points per pixel. Finally, we use a publicly available Stable Diffusion inpainting model (`https://huggingface.co/runwayml/stable-diffusion-inpainting`) to inpaint regions that are empty after rasterization. We use $50$ denoising steps as per the defaults.

It takes on average 2.8 seconds to generate a single $256 \times 256$ image with this baseline on a single NVIDIA RTX 3090 GPU.

### A.3 PixelNeRF

For PixelNeRF [22], we use the implementation from the original authors at `https://github.com/sxyu/pixel-nerf`. We use a pretrained model trained on the same datasets as ZeroNVS [8].

It takes on average 5.8 seconds to generate a single $256 \times 256$ with PixelNeRF on a single NVIDIA RTX 3090 GPU.

## B Simulated Experimental Details

Here we provide details regarding the simulated experimental setup. As a high level goal, we aim to minimize differences from our setup from existing robotic learning pipelines to demonstrate how this augmentation technique can be generally and easily applied across setups.

### B.1 Simulation Environments and Datasets

Our simulated experiments use environments created in the MuJoCo simulator and packaged by the `robosuite` [44] and MimicGen [40] frameworks.

For the Lift, PickPlaceCan, and Nut Assembly tasks, the training datasets are the Proficient-Human datasets for those tasks from `robomimic` [37] and consist of 200 expert demonstrations each. For all MimicGen tasks (Threading, Hammer Cleanup, Coffee, Stack) the datasets consist of the first

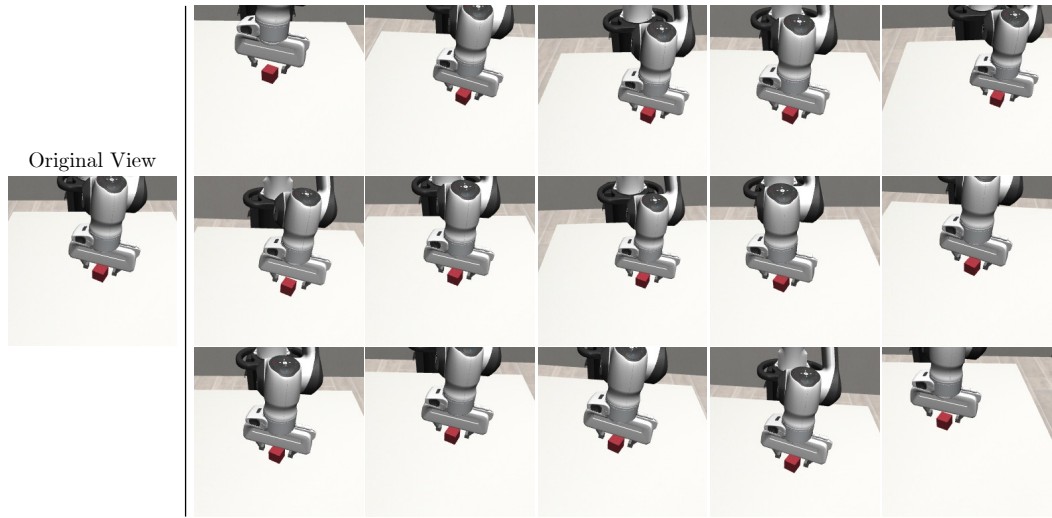

Figure 7: Example ground truth viewpoints from the "perturbation" distribution for the Lift task, rendered by the simulator.

200 expert trajectories for the "core" MimicGen-generated datasets, downloaded from https://github.com/NVlabs/mimicgen_environments.

## B.2 Details for Training and Test Viewpoints

We use the same distribution of viewpoints at training time for augmenting the dataset and when testing the policies. Note, however, that images generated by novel view synthesis models are not guaranteed to actually be from the target viewpoint – only the oracle that uses the simulator to render the scene satisfies this.

Due to the lack of widely adopted testing settings for testing robotic policies on novel views and that the effect of a particular view distribution is highly environment dependent, the hyperparameters for the view distribution were selected by hand by the authors to approximate reasonable distributions that a robot learning practitioner may encounter in practice. We hope these distributions may also be reasonable testing settings for evaluating future methods on these tasks.

### B.2.1 Perturbations

This set of viewpoints are representative of incremental changes, for instance, that of a physical camera drifting over time or subject to unintentional physical disturbance. Specifically, this distribution is parameterized by a random translation $\Delta t \sim \mathcal{N}(0, \mathrm{diag}(\sigma_t^2))$ and rotation around a uniformly randomly sampled 3D axis, where the magnitude is sampled from $\mathcal{N}(0, \sigma_r{}^2)$. Specifically, we set $\sigma_t = 0.03$ m and $\sigma_r = 0.075$ rad. Samples of observations taken from viewpoints drawn from this distribution are shown in Figure 7.

### B.2.2 Quarter Circle Arc

This is a more challenging distribution of camera poses with a real-world analogue to constructing another view of a given scene. We first compute a sphere centered at the robot base and containing the initial camera pose. We then sample camera poses on the sphere at the same $z$ height and within a 90° azimuthal angle of the starting viewpoint. The radius of the sphere is further randomly perturbed with Gaussian noise with variance $\sigma_r^2$. Specifically, the radius of the sphere is 0.7106 m for all simulated environments, which is the distance between the camera and the robot base in the Lift task, and $\sigma_r = 0.05$ m.

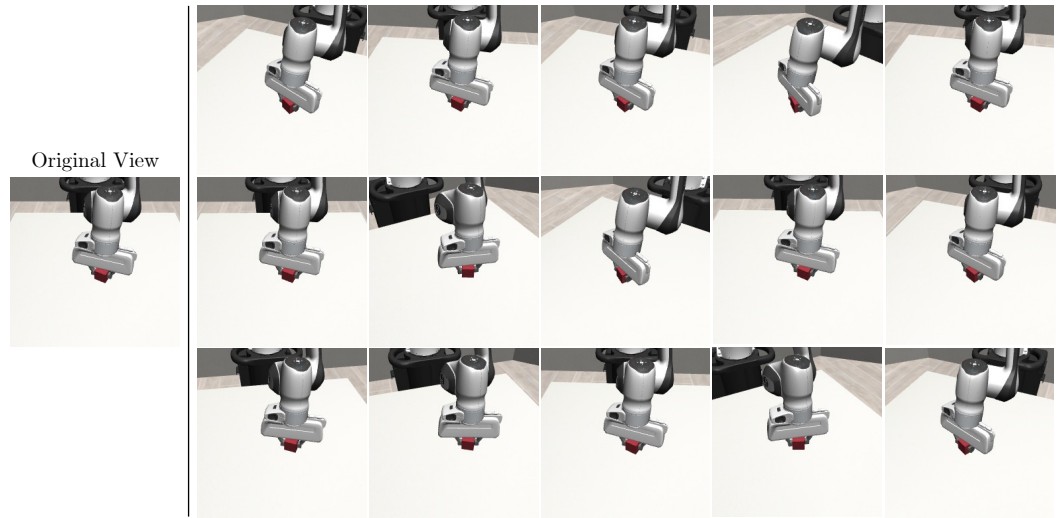

Figure 8: Example ground truth viewpoints from the "quarter circle arc" distribution for the Lift task, rendered by the simulator.

### B.3 Finetuning ZeroNVS on MimicGen Datasets

We finetune the ZeroNVS model on datasets from the MimicGen data of 8 tasks: `stack three`, `square`, `three piece assembly`, `mug cleanup`, `pick place can`, `nut assembly`, `kitchen`, and `coffee prep`. For each environment, we take the first 200 trajectories of the "core" Mimic-Gen dataset for that task with the maximum initialization diversity (e.g. if `Square` is available in variants D0, D1, and D2, we take D2) and simulate 10 random viewpoints from the quarter circle arc distribution for each image in the dataset. We supply this as training data to ZeroNVS, using the training settings from the original ZeroNVS paper but changing the optimizer from AdamW to Adam and decreasing the learning rate to 2.5e-5, and decreasing the batch size to 512 due to computational constraints. We finetune the model for 5000 steps using four NVIDIA L40S GPUs. This takes approximately 16 hours of wall clock time.

### B.4 Policy Learning

We use the same policy training settings for all simulated experiments, taken from the behavior cloning implementation in `robomimic`. The output of the policy network is a Gaussian mixture model. A brief overview of hyperparameters, corresponding directly to `robomimic` configuration file keys, are listed in Table 4. Note that we do not tune these hyperparameters and simply use them as sensible defaults. We train each policy using a single NVIDIA TITAN RTX GPU.

Because we generate the augmented dataset prior to performing policy learning, the computational cost of training policies is split between the augmented dataset generation and policy learning. In our experiments these take relatively similar time durations (around 10-20 hours for dataset generation and 15 for policy learning, varying slightly on the task), however, to achieve this we perform data augmentation parallelized across 10 GPUs. This roughly doubles the total wall clock time required to train the policies.

## C Real World Experimental Details

Next we provide details regarding the real world experimental setup. As a high level goal, we aim to minimize the differences from existing robotic learning pipelines to demonstrate how this augmentation technique can be generally and easily applied across setups.

| Hyperparameter | Value |
| --- | --- |
| Batch size | 16 |
| Optimizer steps per epoch | 500 |
| Training epochs | 600 |
| Input image resolution | $84 \times 84$ |
| Augmentation | Random crop ($84 \times 84 \rightarrow 76 \times 76$) |
| Optimizer | Adam |
| Learning rate | 1e-4 |
| Actor layer dimensions | 1024, 1024 |
| GMM num modes | 5 |
| GMM min std | 0.0001 |
| GMM std activation | softplus |
| Visual encoder backbone | Resnet18 |
| Visual encoder feature dim | 64 |
| Visual encoder pooling | Spatial softmax |
| Spatial softmax num kp | 32 |
| Spatial softmax temperature | 1.0 |

Table 4: Behavior cloning hyperparameters for simulated experiments.

## C.1 Real World Robot Setup

We use a Franka Research 3 (FR3) robot in our real world experiments. The hardware setup is otherwise a replica of that introduced by Khazatsky et al. [7]. Specifically, the robot is mounted to a mobile desk base (although we keep it fixed in our experiments) and two ZED 2 stereo cameras provide observations for the robot. An overview of the real-world robot setup is shown in Figure 9.

We use a Meta Quest 2 controller (also as per the DROID hardware setup) to collect teleoperated expert demonstrations. We collect 150 human expert demonstrations of the `place cup on saucer` task, randomizing the position of the cup and saucer after each task. Each demonstration trajectory lasts approximately 15 seconds of wall clock time.

When performing evaluations, we score task completion based on two stages: 1) **R**eaching the cup in a grasp attempt based on determination by a human rater and 2) **C**ompleting the task, which means that the cup is above and touching the surface of the saucer at some point during the trajectory.

## C.2 Finetuning
## ZeroNVS on the DROID Dataset

To finetune ZeroNVS on the DROID dataset, we first collect a random subset of 3000 trajectories from the DROID dataset. Then, for each trajectory, we uniformly randomly sample 10 timestamps from the du-

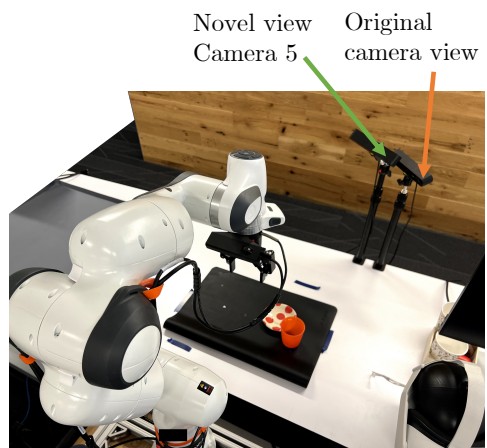

Figure 9: Experimental setup for real robot evaluation. Here we show the testing setup for one particular novel camera view, camera 5. The original camera view that data was collected using is shown by the orange arrow. We use the left camera of each stereo pair.

ration of the video, and consider the trajectory frozen at each of those times as a "scene". Thus, we effectively have 30000 scenes. For each scene, we extract 4 views, which correspond to stereo images from the two external cameras. Although the DROID dataset does contain wrist camera data, we do not use it, as the wrist camera poses are much more challenging for synthesizing novel views.

We then perform depth estimation for each image using a stereo depth model. We then center crop all images to be square, and resize them to $256 \times 256$ to fit the existing ZeroNVS models. We obtain

camera extrinisics from the DROID dataset, and use simplified intrinsics assuming a camera FOV of 68 degrees for all cameras, which we obtained from a single randomly sampled camera in the dataset. In reality, the FOV differs slightly for each camera due to hardware differences, and slightly better results may be obtained by using per-camera intrinsics.

As in the simulated finetuning experiments, we again use the training settings from the original ZeroNVS paper but change the optimizer from AdamW to Adam and decrease the learning rate to 2.5e-5, and decrease the batch size to 512 due to computational constraints. We use 29000 scenes for training and 1000 for validation. As an attempt to reduce overfitting, we mix in a single shard of 50 scenes each from the CO3D and ACID datasets which are sampled for each training sample with probability 0.025 each. DROID data is sampled with probability 0.95. We did not extensively validate the effect of this data mixing due to computational cost of finetuning the model repeatedly, and it is likely unnecessary. We finetune the model for 14500 steps using four NVIDIA L40S GPUs. This takes approximately 50 hours of wall clock time.

### C.3 Policy Learning

**Training augmentation viewpoints.** For the real world experiments, we do not have access to the test viewpoint distribution. To sample viewpoints for ZeroNVS data augmentations for these experiments, we sample from a distribution parameterized in the same way as the "perturbation" range in the simulated experiments, but with a vastly increased variance in translation and rotation magnitude intending to cover a wide range of possible test viewpoints.

This distribution is parameterized by a random translation $\Delta t \sim \mathcal{N}(0, \text{diag}(\sigma_t^2))$ and rotation around a uniformly randomly sampled 3D axis, where the magnitude is sampled from $\mathcal{N}(0, \sigma_r^2)$. Specifically, we set $\sigma_t = 0.15$ m and $\sigma_r = 0.375$ rad.

**Policy learning.** For policy learning on the real robot, we train diffusion policies [38]. Specifically, we use the implementation from the evaluation in the DROID paper [7] with language conditioning removed. The input images are of size $128 \times 128$, and both random color jitter and random crops (to $116 \times 116$) are applied to the images during training. We train all policies for 100 epochs (50000 gradient steps), using 2 NVIDIA RTX 3090 or RTX A5000 GPUs.

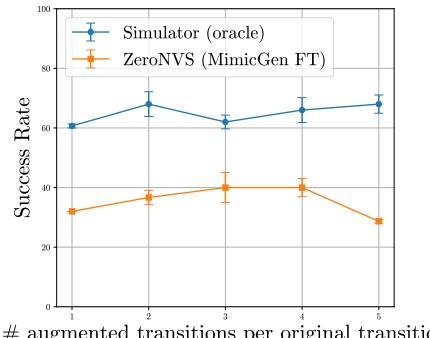

### D Additional Experiments

#### D.1 Increasing Number of Augmented Transitions

In the experiments presented in Section 5, we perform data augmentation via novel view synthesis by doing offline preprocessing of the dataset, augmenting and replacing each transition with a single augmented transition. However, many random data augmentation strategies for neural network training perform augmentation "on-the-fly", applying augmentations to each particular batch. This increases the effective dataset size. Augmentation with novel view synthesis methods is too computationally expensive to apply per batch with our computational budget, but we are still interested in understanding how the performance of trained policies is affected by increasing the number of augmented trajectories for each original dataset trajectory.

Figure 10: Results for ablation of number of augmented transitions per original dataset transitions. Overall, we see modest performance improvements from increasing the amount of augmented data, across both the oracle and learned NVS model.

For the `threading` task with viewpoints sampled from the "quarter circle arc" distribution, we trained policies on dataset containing 1, 2, 3, 4, and 5 augmented transitions per dataset transition for the simulator (oracle) and ZeroNVS (MimicGen finetuned) models.

| Method | MV-MAE | ZeroNVS (MimicGen) (ours) |
|---|---|---|
| Threading – original viewpoint | $28.7 \pm 2.9$ | $\mathbf{64.7 \pm 5.7}$ |
| Threading – novel viewpoints | $2.0 \pm 1.2$ | $\mathbf{32.0 \pm 0.0}$ |
| Stack – original viewpoint | $38.7 \pm 5.3$ | $\mathbf{80.7 \pm 3.3}$ |
| Stack – novel viewpoints | $6.0 \pm 1.2$ | $\mathbf{62.0 \pm 3.1}$ |
| Can – original viewpoint | $19.3 \pm 7.0$ | $\mathbf{86.0 \pm 3.1}$ |
| Can – novel viewpoints | $4.7 \pm 0.7$ | $\mathbf{40.7 \pm 3.5}$ |
| Coffee – original viewpoint | $44.0 \pm 4.2$ | $\mathbf{88.0 \pm 4.2}$ |
| Coffee – novel viewpoints | $3.3 \pm 0.7$ | $\mathbf{32.7 \pm 2.4}$ |
| Hammer – original viewpoint | $89.3 \pm 3.7$ | $\mathbf{100.0 \pm 0.0}$ |
| Hammer – novel viewpoints | $7.3 \pm 0.7$ | $\mathbf{52.0 \pm 3.5}$ |
| Square – original viewpoint | $9.3 \pm 3.5$ | $\mathbf{63.3 \pm 2.4}$ |
| Square – novel viewpoints | $2.7 \pm 0.7$ | $\mathbf{26.0 \pm 2.0}$ |

Table 5: **Comparison to training using pre-trained multi-view masked autoencoder.** Policy success rates on randomized test viewpoints as percentages and standard error of the mean (SEM) over 3 random seeds. We report the maximum performance across training checkpoints, evaluating for 50 trials following Mandlekar et al. [37].

The results are shown in Figure 10. We find that increasing the number of augmented transitions per original dataset transition yields modest improvements with both models, although there is a surprising dip in performance when using 5 augmented transitions for the ZeroNVS (MimicGen finetuned) model.

### D.2 Multi-View Masked World Models Comparison

Here we conduct a baseline comparison to the multi-view masked autoencoding (MV-MAE) method from Multi-View Masked World Models (MV-MWM) [6]. While MV-MWM has a similar motivation to our work, it has a very different problem setting compared to ours: they assume training-time access to a multi-view dataset, while we assume only access to an offline dataset of trajectories captured from a single viewpoint. Thus we adapt the imitation learning approach described in the MV-MWM work to our finetuning experimental setup. Specifically, we train a MV-MAE, using all hyperparameters from Seo et al. [6] on the finetuning dataset from Experiment Q2. Of particular note, we use a higher rendering resolution for this baseline ($96 \times 96$ compared to $84 \times 84$ for policies trained using our novel view augmentation) to match the image resolution used by Seo et al. [6]. We train for 5000 steps, performing early stopping as we observe overfitting by monitoring the validation loss on datasets for the `coffee`, `hammer`, `stack`, and `threading` tasks. We then freeze the pre-trained encoder and use it to train policies on single-view datasets, testing on the quarter circle arc test view distribution. We show the results in Table 5. Our approach significantly outperforms the multi-view masked autoencoding method.

## E  Additional Qualitative Results

### E.1  Real World Novel View Synthesis

In Figure 11, we provide additional qualitative results of novel views synthesized for the real world `cup on saucer` task. We show synthesized images for views sampled from the training view distribution described in Appendix C.3.

### E.2  Saliency Analysis of Learned Policies

To understand how different combinations of observation viewpoints qualitatively affect learned policies, we additionally conduct an analysis of saliency maps of learned policies trained on third-person views only compared to combined third-person and wrist cameras.

Specifically, we visualize saliency maps of convolutional layers of both simulated and real-world policies using GradCAM++ [45, 46] in Figure 12. We find that wrist camera observations tend to have salient features at locations corresponding to objects nearby or underneath the gripper. Policies with only third-person camera views as input tend to have more salient features corresponding to the robotic arm or gripper itself.

For GradCAM++, we choose the target layer to be the common choice [46] of the last convolutional layer in ResNet18 or ResNet50 for simulated and real policies respectively. For the `threading` policies the target model output is the mean of the output Gaussian mixture model action distribution with the highest probability of being selected (largest logit). For the real-world `cup on saucer` policies the target model output is the mean of the output denoised action sequence. We visualize saliency maps on data from viewpoints sampled from the same test distributions as in **Experiment Q2** (quarter-circle arc) and **Experiment Q4** respectively.

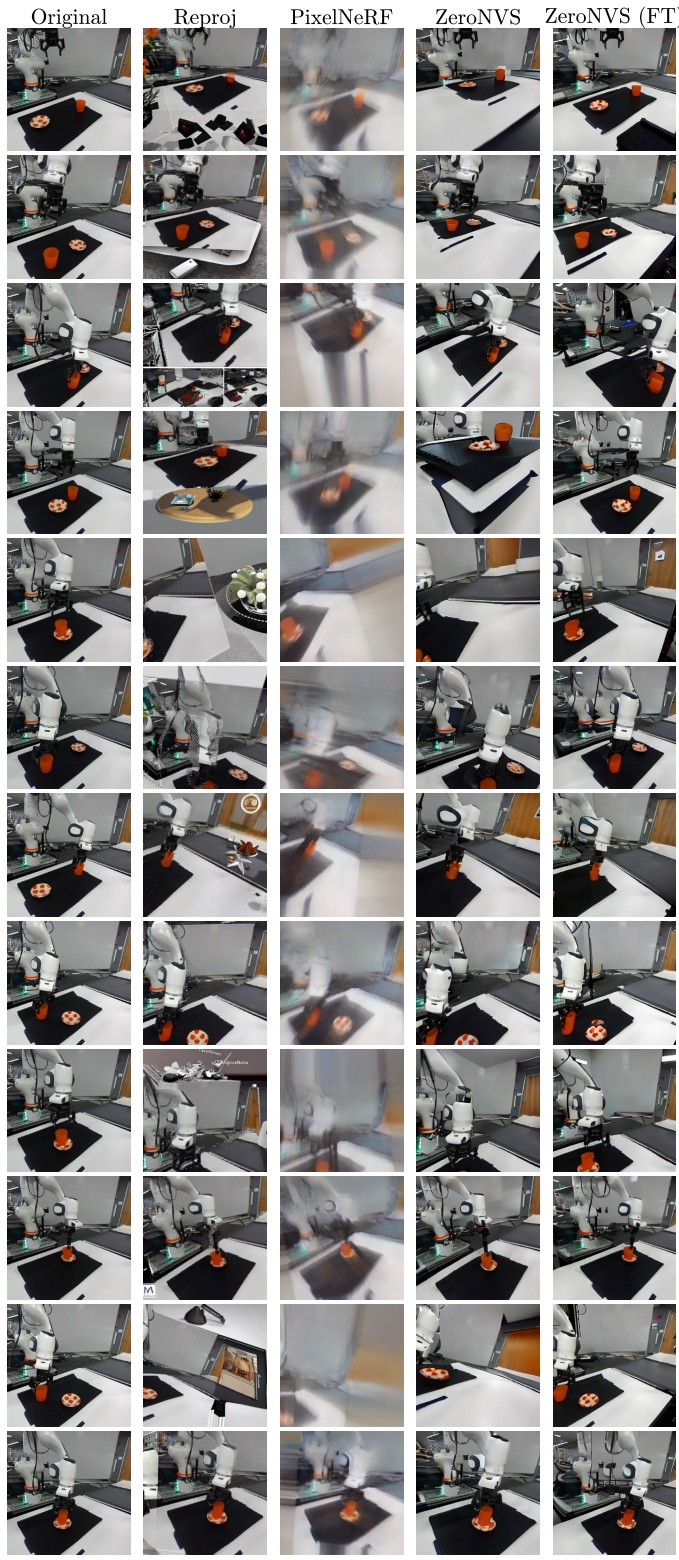

Figure 11: Additional view synthesis results on real robot datasets. The column labeled "Original" is the input view, and random poses are sampled to render the images in the other columns. Note that the ZeroNVS model finetuned on DROID data (rightmost column) is consistently able to generate more crisp, realistic images than the other models, particularly with the robotic arm's visual appearance.

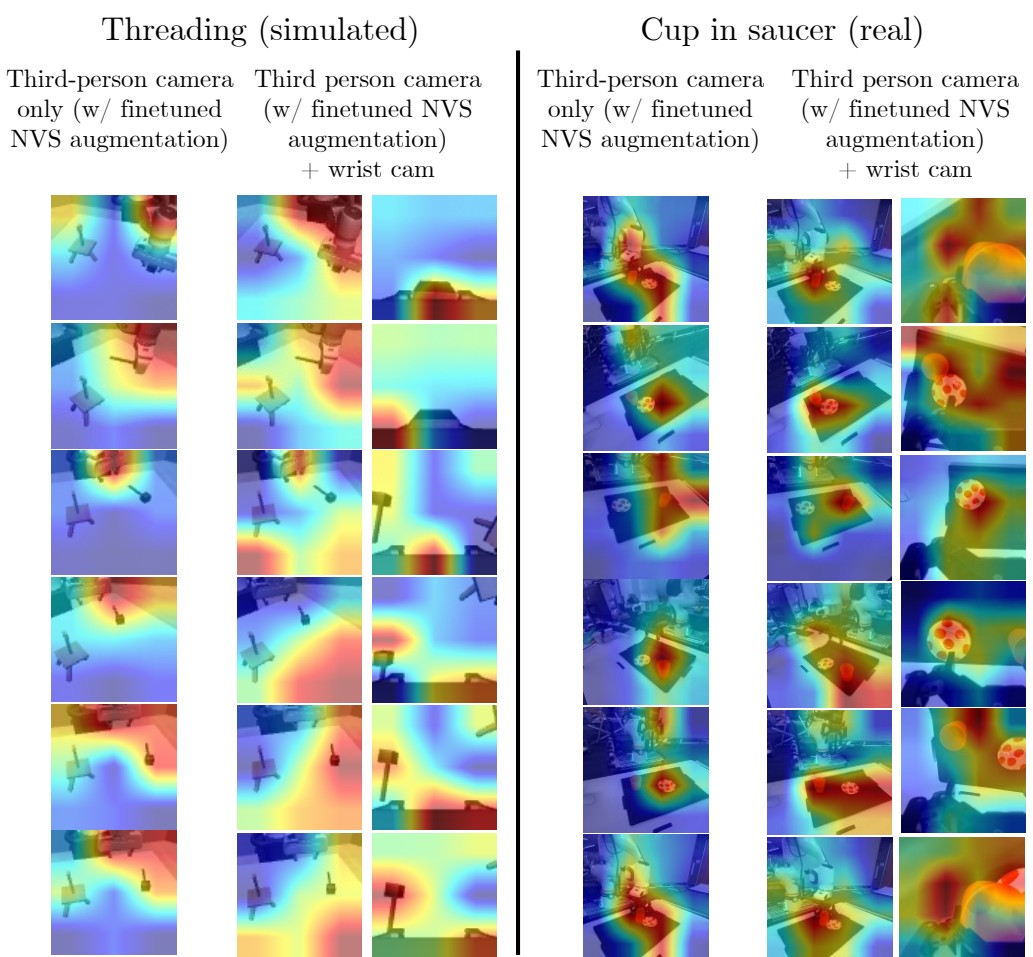

**Threading (simulated)**

Third-person camera only (w/ finetuned NVS augmentation)

Third person camera (w/ finetuned NVS augmentation) + wrist cam

**Cup in saucer (real)**

Third-person camera only (w/ finetuned NVS augmentation)

Third person camera (w/ finetuned NVS augmentation) + wrist cam

Figure 12: Saliency maps computed using GradCAM++ for policies with either solely third-person input views (using augmentation from the ZeroNVS view synthesis model finetuned on MimicGen and DROID data respectively) or third-person and wrist camera viewpoints combined. When incorporated, wrist camera observations do often make contributions to the robot's final action by helping it localize objects grasped by or underneath the gripper. Meanwhile, the policy with only third-person camera views tends to have more salient features corresponding to the robotic arm or gripper itself.

