# OpenReview forum: "View-Invariant Policy Learning via Zero-Shot Novel View Synthesis"
_robot-learning.org/CoRL/2024/Conference — CoRL 2024_

### Official Review · Reviewer_EcF3 · 2024-07-15

**Originality:** 3
**Technical Quality:** 3
**Clarity Of Presentation:** 3
**Potential Impact:** 3
**Recommendation:** 3
**Confidence:** 4

**Review:**

Strength:

- Studies an important problem (adaptation ability of visuomotor policies to novel camera pose) and introduction motivates the problem well.

- Great to see real robot results supporting the simulation results!

- Experiment results show meaningful improvements in performance compared to baselines chosen in this paper.


Weakness:

- I think the algorithm presented in this paper is rather vague and unclear. To let the author easily get the whole pipeline of the proposed algorithm, providng clear pseudocode of it will be of great help.

- The baseline methods chosen in this paper is not strong enough. I would like to see the performance comparison of the proposed method with Multi-View Masked World Model (MV-MWM).

- The presentation of the Zero-Shot Novel View Synthesis in section 3.2 is too brief. ZeroNVS is the major technical component of the proposed method, and not everybody in the robotics community is familiar with ZeroNVS. I would like to see more details that reveal the mechanism of the ZeroNVS. How it works? Why it works better than the other baselines such as PixelNeRF?

**Quality Of The Limitations Section:**

3

**Questions For Rebuttal:**

Please see the list of weaknesses in the section above.

**Robotics Focus:**

4

**Summary Of Paper:**

This paper proposes a method to learn policies robust to changes in camera pose between training and deployment time.  A diffusion-based novel view synthesis model,  ZeroNVS, is used for generating images of the same scene from alternate camera viewpoints given a single input image. Then they replace the original images in the demonstration trajectories with the generated images of the novel view in the training process. Therefore, the trained policy can be applied to novel visual configurations in a zero-shot manner. Experiments demonstrate good zero-shot transfer performance to novel camera views in both the simulation and the real world.

**Summary Of Recommendation:**

The paper is well motivated with a good set of experiments in both the simulation and the real world. Still experiments have room for growth (can add a stronger baseline for comparison) and writing has room for improvement (can have a better presentation of the proposed algorithm).

---

### Official Review · Reviewer_9hYx · 2024-07-20
**Submission 512 Review**

**Originality:** 2
**Technical Quality:** 4
**Clarity Of Presentation:** 4
**Potential Impact:** 3
**Recommendation:** 3
**Confidence:** 4

**Review:**

The paper is very-well written with a clearly-stated hypotheses - i.e. can we use 3D generative models to generate novel views of the same scene and how useful is that for training more robust robot policies. Both hypotheses are tested with sufficient experiments and results are presented and discussed extensively.

Strengths:
* Very high level of clarity.
* Experiments are done for both simulated and real-world robots.
* The problem for making robot learning more robust is very relevant to the field
* Results are demonstrated in video form too - further eases the reader.
* Extensive list of method's limitations.

Weaknesses (some of these are discussed but still):
* Some of the hyper-parameter values - e.g. for the novel view distribution - are not explained.
* No code is mentioned - any plans to release it later?
* It is not quite clear how do we involve the wrist-mounted cameras in the overall paper narrative. Yes, its yet another view of the scene, demonstrating that the more views the better but it doesn't have much to do with the overall thesis which revolves around *view synthesis* with generative models and can thus be somewhat distracting from the main story (as per the title).

**Quality Of The Limitations Section:**

3

**Questions For Rebuttal:**

1) Can you give some more details around the diffusion models used - how easy is it to sample them, fine-tune them (how much resources are required) and how easy is it to add them to the training process. I.e. is training policies with your method 2-3 times slower, for example (or more/less)?
2) Table 3 has missing entries and it's unclear why - can you comment? E.g. for wrist-only model we have dashes everywhere and then 16/20 and 5/20 at the end.
3) The distribution of camera parameters for sampling new views - how do we arrive at those practically. Is there some way of procedurally knowing when we go outside of the distribution of camera parameters that the generative model supports? You mention LPIPS but that is going to be high even if we take the same picture and do affine transforms on it let alone render a new view. Maybe some other metrics from the generative models literature that can give a sense of how real the novel views are - e.g. https://arxiv.org/abs/2002.09797

**Robotics Focus:**

4

**Summary Of Paper:**

This work leverages generative models with 3D priors in order to generate novel viewpoints for a given robot scene in a 0-shot fashion. As a result, a given dataset with demonstrations can be augmented and the policies trained on it can exhibit view-invariance. The authors preset experiments both in simulation and on real robots.

**Summary Of Recommendation:**

I recomment weak accept and am happy to update to strong accept depending on the discussion and answers.

---

### Official Review · Reviewer_Du3J · 2024-07-22
**Simple Idea for data augmentation that show promising directions for view invariant learning**

**Originality:** 2
**Technical Quality:** 3
**Clarity Of Presentation:** 4
**Potential Impact:** 3
**Recommendation:** 3
**Confidence:** 3

**Review:**

The authors propose an approach for using Novel View Synthesis to perform data augmentation to improve viewpoint invariant Imitation Learning to new views during execution time. The authors use ZeroNVS, a zero-shot novel view synthesis approach that they investigate with and without fine-tuning to understand its performance on downstream tasks. Specifically, the authors augment the dataset with images rendered from novel views that are then used to train an Imitation Learning policy on the downstream task. The proposed approach achieves a higher performance on testing from unseen viewpoints which shows the efficacy of the approach both in simulation and on real robot experiments. Additionally, the authors also show the incorporation of wrist camera images to (slightly) improve the generalization capabilities.

Drawbacks:
- A lack of results of the view synthesis on the real robot would help make the results better to understand.
- A better analysis of the wrist camera improvements, would make it better to understand how it affects the task, for example via attention/saliency/convolutional maps. Especially on the real robot setting.
- There is no talk about consistency of a given view along a trajectory. It would be important to discuss how to ensure consistent systhesis along the trajectory for different views.

**Quality Of The Limitations Section:**

3

**Questions For Rebuttal:**

- Have the NVS trained along with the policy learning to learn some task-oriented representations, Or alternatively, any multiview similarity-based policies rather than generating augmentation data from scratch.
- How do the results of Table 3 look like without the wrist camera augmentation? From the results of Q3, while there is a general trend of increased performance due to the wrist camera, it does not seem significantly better.
- Does NVS help improve cross embodiment generalization, especially with the wrist camera view

**Robotics Focus:**

4

**Summary Of Paper:**

Propose a data augmentation scheme for improving generalization to novel view points for Imitation Learning Policies

**Summary Of Recommendation:**

Simple Idea, nice execution, leads to some performance improvement

---

### Author Rebuttal · Authors · 2024-08-09

Thank you to each of the reviewers and the meta-reviewer for your comments and thoughtful reviews that have helped us to improve the paper. We are glad that reviewers found the problem important and well-motivated (9hYx, EcF3), presentation of the work clear (Du3J, 9hYx), and simulated and real-world experiments comprehensive (9hYx, EcF3) with meaningful performance improvements (EcF3). We uploaded an updated manuscript with changes denoted in blue. Here we summarize the major updates and responses to shared questions:
- **Additional comparison to multi-view masked autoencoding (MV-MAE) from Multi-View Masked World Models (MV-MWM) (Seo et al. 2023) (Du3J, EcF3):** We have conducted an additional comparison to the MV-MAEs from MV-MWM by pre-training a masked autoencoder using a multi-view dataset to reconstruct novel views given only a small percentage of unmasked tokens. Specifically, we adapt the imitation learning approach described in the MV-MWM work to our finetuning experimental setup. The results indicate that our approach significantly outperforms the multi-view masked autoencoding representation learning method. The results and implementation details have been added to Appendix D.2.
- **Saliency analysis of policies trained with third-person and wrist camera views (Du3J):** We have conducted additional analysis of policies that use both wrist camera views and third person observations by computing saliency maps for policies trained on real robot data and policies trained on simulated data. The qualitative results are presented in Figure 12 of the revised. We find that wrist camera observations tend to have salient features at locations corresponding to objects nearby or underneath the gripper. Policies with only third-person camera views as input tend to have more salient features corresponding to the robotic arm or gripper itself.
- **Additional details about the ZeroNVS novel view synthesis model (9hYx, EcF3):** We have added additional descriptions of the mechanisms of the ZeroNVS model to make the paper more accessible to readers in Section 4, and practical details to better contextualize our method’s computational costs in Appendix A/B/C.
- **Code release (9hYx):** We clarify that we will release all code and weights upon the paper’s publication.
- **Algorithm pseudocode (EcF3):** We have added pseudocode for the proposed method (see Algorithm 1) to illustrate the entire pipeline clearly.

Thank you for your valuable feedback on our work!

---

### Decision · Program_Chairs · 2024-09-04

**Decision:**

Accept

**Comment:**

The reviewers found some strengths in this submission, but also clearly articulated some questions for the rebuttal phase.
Thank you for your responses.